# Wearable Devices for Ergonomics: A Systematic Literature Review

**DOI:** 10.3390/s21030777

**Published:** 2021-01-24

**Authors:** Elena Stefana, Filippo Marciano, Diana Rossi, Paola Cocca, Giuseppe Tomasoni

**Affiliations:** Department of Mechanical and Industrial Engineering, University of Brescia, via Branze 38, 25123 Brescia, Italy; filippo.marciano@unibs.it (F.M.); paola.cocca@unibs.it (P.C.); giuseppe.tomasoni@unibs.it (G.T.)

**Keywords:** wearable technology, human factors, sensor, work-related musculoskeletal disorder, biomechanical risk, risk factor, real-time measurement, Industry 4.0

## Abstract

Wearable devices are pervasive solutions for increasing work efficiency, improving workers’ well-being, and creating interactions between users and the environment anytime and anywhere. Although several studies on their use in various fields have been performed, there are no systematic reviews on their utilisation in ergonomics. Therefore, we conducted a systematic review to identify wearable devices proposed in the scientific literature for ergonomic purposes and analyse how they can support the improvement of ergonomic conditions. Twenty-eight papers were retrieved and analysed thanks to eleven comparison dimensions related to ergonomic factors, purposes, and criteria, populations, application and validation. The majority of the available devices are sensor systems composed of different types and numbers of sensors located in diverse body parts. These solutions also represent the technology most frequently employed for monitoring and reducing the risk of awkward postures. In addition, smartwatches, body-mounted smartphones, insole pressure systems, and vibrotactile feedback interfaces have been developed for evaluating and/or controlling physical loads or postures. The main results and the defined framework of analysis provide an overview of the state of the art of smart wearables in ergonomics, support the selection of the most suitable ones in industrial and non-industrial settings, and suggest future research directions.

## 1. Introduction

Wearable devices constitute an emerging approach [1], and are excellent candidates for supporting human activities and improving quality of life [2]. They represent a new means of addressing the needs of many industries [3], and have the potential to increase work efficiency among employees, improve workers’ physical well-being, and reduce work-related injuries [4]. Wearable technology extends our capabilities as humans, and epitomises the interaction of humans and technology [5]. An industrial wearable system supports real-time, trusting, and dynamic interaction among operators, machines, and production systems, providing a human-centric empowering technology in Industry 4.0 [6]. Compared to computers and mobile phones, it can provide many different ways of human computer interaction for users to strengthen their experience [7].

A wearable device is essentially a tiny package with powerful sensing, processing, storage, and communications capabilities [5], and the term can refer to “any electronic device or product designed to provide a specific service that can be worn by the user” [8]. Other definitions underpin its ability to create interaction between users and the smart environment anytime and anywhere [9], and to measure information such as the users’ locations, environments, movements, and vital signs [10]. The capabilities of these devices to measure various physiological and kinematic parameters, assess human performance, monitor human movement, perform motion analysis in a real manufacturing scenario, and/or record user’s kinetics, kinematics, physical parameters and/or (psycho)physiological parameters are also emphasised by other authors in the literature (e.g., [11,12,13]).

Wearable solutions can have various forms and are very different in their application [14]. In the literature, several classifications of wearable devices are available, and a standard one has not yet been provided [15]. Interesting different taxonomies can be found in Mardonova and Choi [3], Khakurel et al. [4], Mewara et al. [7], and Park and Jayaraman [16]. Among these possible categorisations, Dimou et al. [17] consider the field of use of smart wearables, such as lifestyle, entertainment, medical, fitness, gaming, and industrial. This classification highlights the pervasive use of such technology in different fields, as pointed out by several researchers in the literature (e.g., [7,9,15,16,18]).

Regardless of the different types of wearable equipment, the technology shares certain common features and attributes. A wearable device should be used while the wearer is in motion, be not merely attached to the body but become an integral part of the person’s clothing, allow the user to maintain control, and be constantly available [7]. It should be lightweight, aesthetically pleasing, shape-conformable, multi-functional, and easily configurable for the desired end-use application [16]. Additionally, it should improve the body condition, comfort, and safety of his/her wearer and, possibly, of surrounding or distant people, and its design should be gender and culture—oriented [2].

The wide spectrum of wearable categories, potentials, and applications has inspired the conduction of several researches and many reviews of their use in various fields [9]. For instance, Seneviratne et al. [18] survey the trends, technologies, research challenges, and solutions for commercially available wearable devices and research prototypes. Khakurel et al. [4] systematically review the trend of wearable technology to assess both its potential in the work environment and the challenges concerning its utilisation in the workplace. Mardonova and Choi [3] review trends in wearable device technology, providing an overview of its prevalent and potential applications to the mining industry. Also recent studies give insights of the state of the art of wearable equipment: e.g., Chander et al. [11] focus on wearable stretch/strain sensors technology for human movement monitoring and fall detection, Koutromanos and Kazakou [9] on the use of smart wearables in primary and secondary education, and their impact on learning and teaching, and Niknejad et al. [15] on recent advances and future challenges of smart wearables. Also in the ergonomic field some reviews have been published, but considering only a limited scope of the wearable device use: Tsao et al. [19] summarise the applications of wearable sensors for human work and status evaluation, whereas Lim and D’Souza [20] synthesise the literature on body-worn inertial sensing for assessing biomechanical exposures and musculoskeletal disorder risk resulting from physical work.

The adoption of wearable technology appears particularly interesting for ergonomic purposes because of the well-known properties of assisting the users anywhere [21], sensing, collecting, and uploading data in a 24 × 7 manner [18], and monitoring continuously human performance [11]. Ergonomics (or human factors) is the scientific discipline concerned with the understanding of interactions among humans and other elements of a system, and it applies theory, principles, data, and methods to optimise human well-being and overall system performance [22,23]. In particular, this discipline promotes a holistic, human-centred approach to task, product, environment, and system design and evaluation, considering physical, cognitive, organisational, environmental, and other relevant factors [23]. As highlighted by Karwowski [23], the traditional domains of specialisation are the following:Physical ergonomics, which is mainly related to human anatomical, anthropometric, physiological, and biomechanical characteristics as they relate to physical activity.Cognitive ergonomics, which focuses on mental processes (e.g., perception, memory, information processing, reasoning, and motor response), as they affect interactions among humans and other elements of a system.Organisational ergonomics, which is concerned with the optimisation of socio-technical systems, including their organisational structures, policies, and processes.

Therefore, the ergonomics has the purpose to improve the performance of systems by improving human-machine and human-computer interactions [24]. It shall be used in a preventive function by being employed from the beginning, but can be also successfully applied in the redesign of an existing work system [25]. It is recommended that a work system be designed for a broad range of the target population, which is the people for whom the design is intended, specified according to relevant characteristics [22,25]. In such a context, the role of ergonomics is two-fold: the first one is to understand purposive interactions between people and artefacts and especially to consider the capabilities, needs, desires, and limitations of people in such interactions, and the second one comprises a contribution to the design of interacting systems, maximising the capabilities, minimising the limitations, and trying to satisfy the needs and desires of the human race [26].

To the best of our knowledge, there are no systematic reviews on the potential use of wearable devices in ergonomic applications. This represents a gap in the literature that this paper overcomes. Its aim is to conduct a systematic review in the scientific literature to answer the following primary research question: “Which wearable devices have been proposed for ergonomic purposes in the scientific literature?”. In addition, some secondary research questions are defined to assist this literature review:Which ergonomic risk factors are analysed by means of these wearable devices?Which ergonomic purposes are achieved by means of these wearable devices?Which ergonomic criteria are at the basis of the use of these devices?Which populations can benefit from the use of these wearable devices?Are these wearable devices applied and/or validated in real contexts?

The conduction of this systematic review and the answers to all the research questions may support both researchers and practitioners during the design, realisation, and testing of wearable devices for ergonomic purposes. Researchers may consult in a single document an overview of the state of the art of smart wearables in ergonomics available in the literature, whereas practitioners can be guided when selecting the most suitable wearable technology and be stimulated to increase its proper adoption.

The remainder of the paper is organised as follows: Section 2 describes the methods followed in this research. Results are presented in Section 3 and discussed in Section 4. Concluding remarks are provided in the final section.

## 2. Materials and Methods

A systematic review is an explicit and reproducible research methodology to answer one or more specific research questions on a specific topic, identifying all relevant studies and summarising the state of the art [27,28].

The systematic review described in this paper was conducted according to the Preferred Reported Item for Systematic Reviews and Meta-Analyses (PRISMA) statement [29]. This allowed critically identifying, selecting, assessing, and analysing all relevant studies answering our primary and secondary research questions.

### 2.1. Eligibility Criteria

The relevance of each paper retrieved to answer the research questions was assessed based on the following inclusion (or eligibility) criteria:Only papers written in English.Only papers published in scientific journals or conference proceedings.Only papers proposing a wearable device with an explicit ergonomic purpose.Only papers proposing a new wearable device, or the use of an already available device for novel ergonomic reasons not previously addressed.

Concerning the third and fourth inclusion criteria, we defined the following exclusion criteria in order to simplify the study selection and classification of the retrieved papers:Papers proposing a wearable device focusing only on parameters not explicitly related to ergonomics (e.g., only joint angle measurement [30]).Papers proposing a wearable device to consider one or more risk factors, without explicit ergonomic assessments or improvements (e.g., visual load [31]).Papers proposing a wearable device for purposes different from the ergonomic ones (e.g., rehabilitation [32]).Papers proposing a comparison among wearable devices considering only features not correlated with ergonomics (e.g., [33])Papers proposing only the ergonomic evaluation of a wearable device (e.g., [34]).Papers proposing only a qualitative or technical description of a wearable device (e.g., [35]).Papers proposing only a design approach or validation protocol of wearable devices (e.g., [36]).

### 2.2. Search Strategy

The literature search was carried out on four electronic databases, relevant to the fields of interest for our systematic review: IEEEXplore, PubMed, Scopus, and Web of Science. The following search terms, classified in two groups, were used: (1) wearable device, wearable solution, wearable system, wearable technology, wearable equipment, wearable computer, wearable computing, smart wearable; (2) ergonomics, human factors.

The searches in the electronic databases were performed on 31 August 2020. Each database was queried from the date of the oldest indexed paper in order not to exclude potentially relevant studies and analyse the distribution of the papers over time. Title, abstract, and keywords were the fields considered in the queries. The search was limited to English documents.

The search strategy for each electronic database is detailed in Table 1.

### 2.3. Study Selection

Once the databases were queried, we employed the reference management software Endnote^®^ X9.3.3 (Clarivate, Philadelphia, PA, United States) for recording references, removing multiple records, and creating a unique database of references. The subsequent manual removal of other duplicates allowed obtaining a unique library representing our initial database.

To screen the papers in the initial database, we applied a three-stage process based on Stefana et al. [37]: (1) title evaluation, (2) abstract and keywords evaluation, and (3) full-text evaluation. In each stage of the screening process, three authors critically appraised the papers in parallel and independently; all the documents selected by at least one reviewer have been promoted to the successive screening stage to be over-inclusive and minimise the chance to discard relevant papers [38]. In particular, the aim was to exclude irrelevant studies during the stages (1) and (2), and examine the remaining documents on the basis of the above eligibility criteria during the stage (3). At the end, we collected the included studies (i.e., the documents answering the research questions) in the final database and recorded the primary reason for exclusion of the other papers referring to the defined criteria.

The intermediate selection process results and stages followed to obtain the final database are summarised in Figure 1.

The final steps of the systematic review were the characterisation and analysis of the papers in the final database. For such characterisation and analysis, we defined specific comparison dimensions that will be described in detail in the following sections.

## 3. Results

The systematic review returned 28 papers proposing 24 wearable devices for one or more explicit ergonomic purposes. The number of papers is different from the number of wearable devices because some authors have discussed their research studies in more than one paper: Caputo et al. [39,40,41], Conforti et al. [13,42], and Peppoloni et al. [43,44]. The following analysis is based on the 24 studies, each one focusing on a wearable technology.

Although in the search strategy a starting date was not defined, the first relevant paper was published only in 2014 by Peppoloni et al. [43]. The distribution over time of the results, displayed in Figure 2, shows that more than half of the papers were published in 2019 and 2020. Furthermore, even though the review was stopped on 31 August, the majority of articles were published in 2020. This trend confirms a recent and growing interest in this area of research.

The journals that published the highest number of papers are *Applied Ergonomics*, *Automation in Construction*, and *Sensors*, with three papers each. Other journals published one paper each, as shown in Figure 3.

Regarding the conference papers, the proceedings of *International Conference on Human Factors and Wearable Technologies* and *IEEE International Workshop on Metrology for Industry 4.0 and IoT* include two papers each; the former in the same edition, while the latter in different editions.

To analyse the papers constituting the final database and answering our research questions, the following eleven comparison dimensions were defined: (1) type of wearable device, (2) being ready to wear, i.e., the possibility to put the smart wearable easily and readily on the users, (3) parts of the body where the device can be worn, (4) physical, cognitive, organisational factors traditionally considered in ergonomics and studied by means of the wearable technology, (5) ergonomic risk factor analysed through the utilisation of the wearable device, (6) task performed by the population, (7) main purpose and use from the ergonomic perspective, i.e., if the device can be applied for performing assessments and/or obtaining improvements, (8) type of output information provided by the wearable device to the users and its interface, (9) criteria at the basis of ergonomic assessment or improvement, in terms of standards, methods, principles, and/or guidelines, (10) population, i.e., the device users identified in the study, and (11) application and validation of the technology in experimental tests, simulations, and/or real contexts.

Table 2 points out the relationships between these comparison dimensions and our primary and secondary research questions, while Table 3 analyses the 24 studies with respect to such dimensions (in alphabetical order by author). Note that in Table 3 the details related to physical, cognitive, and organisational factors are omitted because all the studies deal with physical ones, none of them consider organisational ones, and only one (i.e., Kunze et al. [45]) evaluates cognitive ones.

### 3.1. Wearable Devices for Ergonomic Purposes

Diverse kinds of wearable devices are proposed for ergonomic purposes by the retrieved studies: two insole pressure systems, fourteen sensor systems, two smart garments, a robot, a smartwatch, a vibrotactile feedback interface, a pair of smart glasses, and two body-mounted smartphones. Therefore, the majority of the studies deal with a sensor system. We use the term “sensor system” taking inspiration from Park and Jayaraman [16]: a sensor system is a platform composing of different types and numbers of sensors, which are positioned in different locations on the body and whose signals are processed in parallel and combined to provide specific parameters in real-time. Since a large number of different types of sensors can be required on various parts of the body, a sensor system is not immediately ready to be worn. Table 4 details the components in terms of Inertial Measurement Units (IMUs) and complementary wearable sensors of the different sensor systems available in the scientific literature (in alphabetical order by author). Regarding the wearable device described by Lins et al. [56], in this table we report only the elements forming the sensor system.

All the sensor systems, with the exception of those described in [50] and [58], are based on IMUs. Such component combines information obtained from multiple electromechanical sensors (e.g., accelerometers, gyroscopes, and magnetometers) [57], and is self-contained and unobtrusive [44]. In accordance with Cerqueira et al. [48], “regarding the positioning of the IMUs on the human body, there is no standard established protocol and each author proposes different locations for the sensors”. Indeed, the analysis of the part of the body on which a sensor system can be worn corroborates the possibility of attaching the different sensors in several body segments, such as the trunk and upper extremities. We identified head, upper extremity, hand, trunk, lower extremity, foot as the relevant parts of the body, mainly based on the International Standard ISO 11226 [65]. Giannini et al. [52] present the only wearable device that can be worn on all the six considered body segments: it is a sensor system composed of 17 IMUs placed on head, sternum, shoulder blades, upper arms, lower arms, hands, pelvis, upper legs, lower legs, and feet thanks to elastic bands and connected to a central unit, which in turn is connected to the battery pack that can last until 9.5 h.

The trunk and upper extremities also represent the part of the body on which other types of wearable devices can be worn: smart garments [48,51] and body-mounted smartphones [60,61]. If a smart garment can be put easily and readily on the users, a body-mounted smartphone is only partially ready to wear. The trunk, which is the most frequent segment covered by smart wearables for ergonomic purposes, is the human location on which also a robot [53] or a vibrotactile feedback interface [21] can be placed. Other portions on which a wearable technology can be put on are foot in the case of insole pressure systems [46,47], hand for a smartwatch [54], or head during the smart glasses use [45]. Those devices are ready to be worn by the users.

### 3.2. Analysed Ergonomic Risk Factors

All the papers consider physical factors. A large number of the studies (16 out of 24) propose wearable devices for analysing unfavourable postures during different types of tasks. In particular, sensor systems represent the most frequent technology employed for such purposes. This confirms that “the real-time assessment of human movements and posture through wearable sensors can inform workers about inadequate lifting postures, dramatically helping in preventing injury risks” [42]. Such results also emphasise the statement by Lu et al. [57], according to which “the application of IMUs for tracking human motion as a part of the ergonomic assessment is becoming popular because the collection of the human body motion does not greatly interrupt with workers’ job performance”.

Besides systems based on sensors, smart garments, a body-mounted smartphone, a robot, and an insole pressure system have been developed for evaluating and facing the adoption of awkward postures. In particular, the smart garment by Cerqueira et al. [48] and the body-mounted smartphone by Nath et al. [61] are proposed for explicitly dealing with tasks requiring awkward postures. This type of task is also investigated by Lins et al. [56] and Yan et al. [64] that describe wearable devices based on sensor systems.

In 3 studies the authors do not evaluate only postures as ergonomic risk factor: Jin et al. [54] explore the biomechanical loads of the neck and shoulder regions in standing and sitting postures during typical activities performed by means of a smartphone (e.g., calling and message checking), Peppoloni et al. [43,44] concentrate their attention on an online assessment of both postures and muscular efforts during repetitive jobs, and Kunze et al. [45] propose the only wearable system in our review that also considers cognitive factors. This contribution provides details related to smart glasses able to detect the too steep head angle and combat computer vision syndrome during reading and talking tasks.

The remaining 8 studies focus mainly on physical loads. Five sensor systems, an insole pressure system, a vibrotactile feedback interface, and a body-mounted smartphone are developed for ergonomic evaluation of this physical factor during tasks involving manual material handling, manual handling of low loads at high frequency, heavy or repetitive tasks, or physically demanding jobs. Manual material handling represents the most investigated task among these studies (4 out of 8), while fitness exercises are taken into account by only one study (Manjarres et al. [58]).

None of the studies retrieved by our systematic review concern organisational factors.

### 3.3. Ergonomic Purposes

The majority of the studies (16 out of 24) propose wearable solutions for assessing ergonomic risk factors. Besides sensor systems, two insole pressure systems [46,47] and two body-mounted smartphones [60,61] are developed for producing post-exposure recognition or assessment of postures, activities, or overexertion risks. Some of them (e.g., [46,60]) employ machine learning techniques for such recognition or assessment, such as a supervised machine learning classifier and a support vector machine learning classifier, by using MATLAB^®^. MATLAB^®^ is a programming and numeric computing environment that is utilised by 9 studies out of 24 for analysing and processing data, and implementing machine learning techniques, algorithms, or computations.

In 3 studies [45,53,54] the main ergonomic purpose regards the improvement of the analysed risk factor. In particular, Hahm and Asada [53] propose a robot to support the user in awkward postures, Jin et al. [54] a smartwatch able to estimate and analyse joint angles and muscle activity, and Kunze et al. [45] a couple of smart glasses for providing real-time feedback to improve the risk factors, blurring or flipping the screen content away from the user.

The other 5 studies permit both assessing and improving ergonomic risk factors. Two of them [48,51] describe smart garments that monitor real-time postures displayed thanks to a Graphical User Interface (GUI), and implement a feedback strategy. The feedback strategy followed by the studies are different: in Cerqueira et al. [48] the biofeedback is provided by vibrotactile motors, while in Ferreira et al. [51] visual feedback is realised through luminous signalling. As an alternative to visual feedback, alarm sounds by means of a smartphone application are proposed for warning users when ergonomically hazardous operational postures and holding time leading to lower back and neck pain are detected [64]. Vibrotactile motors are also employed by another retrieved study (Lins et al. [56]) describing a wearable technology compromising a sensor system and a vibrotactile feedback interface in order to alert users when unfavourable postures are reached.

All these 5 studies both monitoring a risk factor and guiding users towards more ergonomic conditions permit assessing or estimating the risk factor itself or a parameter correlated to it in real time. Besides them, other wearable devices give real-time outputs, such as sensor systems [43,44,50,52,58] and smart glasses [45]. However, most of the studies (13 out of 24) focus on wearable solutions recognising or detecting the risk factor itself or a parameter correlated to it only after the occurrence of the exposure. Among them, only the smartwatch described by Jin et al. [54] is devoted to ergonomic improvement as the main purpose.

### 3.4. Ergonomic Criteria

The ergonomic criteria quoted in the studies are heterogeneous: 13 studies out of 24 adopt only one approach or standard, five report two types of criteria, one mentions three guidelines by National Institute for Occupational Safety and Health (NIOSH), and another one is based on four methods and three standards. In particular, 4 studies out of 24 (Caputo et al. [39,40,41], Antwi-Afari et al. [46], Valero et al. [63], Yan et al. [64]) refer to the International Standard ISO 11226 [65] that concerns the acceptability of static working postures, and the suggestion of ergonomic recommendations for different work tasks performed by adult workers. Other 4 studies described in 5 papers [43,44,48,50,59] propose wearable devices to assess postures on the basis of Rapid Upper Limb Assessment (RULA), i.e., a survey method for investigating work-related upper limb disorders [66]. In addition to the RULA method, Peppoloni et al. [43,44] also employ Strain Index (SI), i.e., a job analysis methodology for risk of distal upper extremity disorders [67]. As emphasised by Peppoloni et al. [44], the RULA and SI methods are explicitly quoted in the ISO 11228-3 [68] for the risk assessment of repetitive task; the former is based on the kinematic assessment, while the latter is mostly affected by the level of effort, ratio of recovery time, and time under effort. Note that the ISO 11228-3 [68] establishes ergonomic recommendations for repetitive work tasks involving the manual handling of low loads at high frequency. The papers retrieved by this systematic literature review that mention this International Standard are Giannini et al. [52] and Lenzi et al. [55]. Both propose a sensor system for assessing the physical load of workers. Other methods listed in the ISO 11228-3 [68] are used by the studies (e.g., Ovako Working Posture Analysis System, Rapid Entire Body Assessment, ACGIH TLV for lifting), but only one [55] employs the Occupational Repetitive Actions (OCRA) Index, which is the preferred method for detailed risk assessments. To assess manual lifting tasks, 3 studies contained in 4 papers [13,42,52,57] take into consideration the revised NIOSH lifting equation, quoting the applications manual NIOSH 94-110 [69] and the International Standard ISO 11228-1 [70].

Three studies [21,51,61] do not mention any international known methodology, but propose specific principles or categories: Nath et al. [61] suggest several posture categories based on measurements of trunk flexion, trunk lateral bend, should flexion, should abduction, and elbow flexion, Ferreira et al. [51] use the principles of ergonomics for monitoring and analysing the sitting position of an individual, and Kim et al. [21] consider the amplitudes of the overloading joint torques for applying vibrotactile stimuli to the body joints.

### 3.5. Populations

The majority of the studies (21 out of 24) propose a wearable device for workers. Among them, 5 [46,47,61,63,64] focus on construction personnel. Construction jobs are among the most ergonomically hazardous occupations and the workers are frequently exposed to numerous physical risk factors leading to work-related musculoskeletal disorders (WMSDs) [46,61]. The ergonomic risk factors in those 5 studies mainly regard awkward postures. To assess them, Nath et al. [61] propose a body-mounted smartphone, while Valero et al. [63] introduce a sensor system to be worn on upper and lower extremities, and trunk. For minimising the workers’ exposures to awkward working postures, Antwi-Afari et al. [46] develop a wearable insole system using foot plantar distribution data. In order to both assess ergonomically hazardous postures and warning the wearer, Yan et al. [64] improve and apply a sensor system based on IMUs.

Other studies provide insights about industrial workers. Specifically, Caputo et al. [39,40,41] develop a motion tracking system in upper body and full body configurations for assessing postures assumed by the workers during typical industrial working activities, while Lins et al. [56] present a system based on sensors and a vibrotactile feedback interface to assess and improve the workers postures. In addition to workers, one study [58] takes into consideration the physical load of athletes for fitness purposes.

### 3.6. Application and Validation

Most of the studies (16 out of 24) examine the applicability of the proposed wearable devices only by means of experimental tests and/or simulations. For example, the insole pressure systems proposed by Antwi-Afari et al. [46,47] for examining different awkward postures or overexertion-related workers’ activities are tested in a simulated laboratory experiment, the two body-mounted smartphones described in Nath et al. [60,61] are evaluated by means of field experiments involving subjects that perform typical real-world activities at their own pace, and the smart garment by Cerqueira et al. [48] is validated in a simulated scenario comprising five general tasks requiring different working postures, and involving five subjects.

A minority of the studies (4 out of 24) demonstrate the utility of the technology in real contexts. For instance, Meltzer et al. [59] evaluate the ergonomic risks considering 53 surgeons representing 12 surgical specialties, Giannini et al. [52] assess their system in the activity of lift-on/lift-off of containers in a port, Lenzi et al. [55] test the developed toolbox in a real context related to large retail chains involving expert operators and real workers, and Kunze et al. [45] show a couple of demonstrations of their technology from reading detection over ergonomics to talking recognition for social interaction tracking.

The remaining 4 studies rely on both real context applications, and experimental tests and/or simulations. They propose sensor systems for assessing and/or improving postures and/or physical loads. Among them, Peppoloni et al. [43,44] propose applications, preliminary validations, and data collection campaigns of a sensor system for assessing muscular efforts and postures of a check-out operator and super-market cashiers during everyday real-life operations, whereas Caputo et al. [39,40,41] present a wearable inertial motion tracking system by means of simulations, data analyses, algorithms, experimental sessions, and several test cases carried out on assembly lines in Fiat Chrysler Automobiles (FCA) to test the system reliability in industrial environments. Manjarres et al. [58] test the reliability of their sensor system-based activity classifier with twenty subjects, and show an athlete’s application to estimate and track the physical workload for push-ups, squatting, and jogging during each daily session, for twenty days. Finally, Yan et al. [64] describe a laboratory test carried out to validate the proposed IMU-based real-time motion warning system, and a field experiment on a construction site in Hong Kong obtaining an improvement of postures by workers during manual material lifting.

## 4. Discussion

The systematic literature review proposed in this paper confirms the current interest in wearable applications for ergonomic purposes. The distribution over time of the results highlights a growing trend of relevant articles, with more than half of the papers published in 2019 and 2020.

The available solutions allow going beyond the methods and tools traditionally employed for ergonomic assessments (e.g., visual observations and video recording [71]) that can be influenced by the assessor’s competencies and/or cannot be objectively conducted [72]. Compared to such methods and tools, wearable devices permit measuring and monitoring parameters of interest in real-time with greater precision and reliability, and thus analysing and assessing ergonomic risk factors in a wide spectrum of scenarios potentially encountered in working environments. This gives the great possibility of facilitating experts’ diagnostics, and preventing and/or reducing WMSDs. Additionally, as underlined by Valero et al. [73], “the evolution of technologies has been driven by not only improvement in measurement accuracy and precision, but also reduction in intrusiveness and enhanced wearability”. In combination with the potential of assessing ergonomic risk factors, some devices also permit combating unhealthy conditions at source. Indeed, smart garments, vibrotactile feedback interfaces, and sensor systems can alert workers about the achievement of dangerous states and consequently guide them towards ergonomic circumstances. Such devices can allow providing immediate biofeedback to workers that leads to healthier, safer working habits and consequent reduction of medical expenses for musculoskeletal-system related injuries [48]. All these potentialities help to optimise the well-being, minimise the limitations, and satisfy the needs of the operators working in the Industry 4.0 era. In the following paragraphs, the answers to our primary and secondary research questions are provided and discussed in detail, also highlighting several gaps in the literature.

### 4.1. Which Wearable Devices Have Been Proposed for Ergonomic Purposes in the Scientific Literature?

The results of this systematic review underline that the large majority of the proposed smart wearables for ergonomic purposes are based on sensor systems composed of different types and numbers of components located in various parts of the body. This wearable technology is probably preferred because designers can select a variable set of sensors with diverse features in order to measure and monitor the parameters under investigation. This aspect also allows controlling and minimising the costs of the device. Furthermore, taking inspiration from Conforti et al. [13], the estimation of risks through sensor systems promotes the design of setups that are not bulky and are suitable for each working activity. However, there is no a standard design principle related to the number of sensors. Indeed, the number of sensors can vary, adapting to the needs of the particular application [63]. Cerqueira et al. [48] underline that regarding wearable designing, the focus is on the use of the minimum number of sensors, not compromising the performance of the system. For instance, the shoulder complex is typically monitored by only one IMU placed in the upper arm, while the neck by another one located in the forehead or vertebrae C4. According to Lu et al. [57], although generally detailed whole-body biomechanical models to track body motion typically require from 13 to 17 IMUs, a configuration based on five IMUs may adequately reconstruct the whole-body posture to discriminate gross movement activities. In addition to the number of sensors, another relevant parameter to be taken into account during the design stage concerns their location. Sensor placement depends on the task being monitored [62], and their location influences the accuracy of captured data [64]. For example, Sedighi Maman et al. [62] conclude that for physical fatigue detection the sensors on wrist, torso, and hip are required, whereas for physical fatigue level prediction, the wrist, hip, and ankle sensors are needed. These authors also suggest that the sensors can be located on only one side of the body if the purpose is to provide a simplified approach for practical implementation in the workplace. The sensors should be attached to the subject’s body by means of elastic straps and fitted tightly to body segments in order to prevent slippage that could cause incorrect recognition of postures and movements [63]. Sensor systems are not ready to be worn and employed by a user, requiring interventions by others to be installed, set, and monitored. On the contrary, smart garments, insole pressure systems, smart glasses, and smartwatches are easily worn and autonomously put into action by a user. It is thus desirable that the sensor systems, after being developed and tested, are integrated into clothes and equipment in order to make them easy to wear and use.

### 4.2. Which Ergonomic Risk Factors Are Analysed by Means of These Wearable Devices?

The studies obtained thanks to our systematic review mainly focus on the analysis of awkward postures adopted by workers during a variety of tasks and activities. This can presumably be justified by the evidence, even recent, of the prevalence on such specific risk factor: according to European Agency for Safety and Health at Work (EU-OSHA), postures and working in awkward positions are one of the main relevant risk factors related to WMSDs in the back, upper limbs, and/or lower limbs [74]. All the studies address ergonomic physical risk factors focusing not only on the aforementioned postures, but some of them also on the physical load during mainly manual material handling or heavy and/or repetitive tasks. Only one also considers a cognitive aspect, and none deal with organisational issues (e.g., participation, cooperation, and environment). This result points out the capability of smart wearables to quantify objectively various parameters (also physiological ones), but the reasonable difficulties and challenges related to capture organisational issues from the ergonomic point of view. However, the development of modern technologies could also help in this direction, trying to identify objective parameters on which to base an assessment of organisational aspects. Additionally, wearable devices developed to quantify factors such as visual or mental load could be used for ergonomic analysis from a cognitive point of view.

### 4.3. Which Ergonomic Purposes Are Achieved by Means of These Wearable Devices?

The ergonomic purposes achieved by means of the wearable devices are assessment and/or improvement of the analysed risk factors. Assessment permits evaluating users’ exposures prospectively, and can be carried out by means of sensor systems, insole pressure systems, or body-mounted smartphones. For this purpose, the development of graphic interfaces (e.g., GUIs) helps monitoring the parameters considered in the assessment and facilitates the identification of the most critical issues. To assure an efficient identification of the most critical parameters, efforts should be addressed to designing interfaces that display a set of relevant information to the users and could be easily adopted without requiring expertise and competencies in programming.

A minority of papers propose wearable devices integrated with a machine learning approach. Machine learning algorithms can allow recognising automatically dangerous situations [13], and their combination with ergonomics can predict the amount of risks that an activity can represent for a subject [58]. Such integration also permits extracting relevant information to support ergonomics-related decision making. Therefore, research on this integration is highly encouraged.

Wearable solutions developed for ergonomic assessments do not provide real time warnings for increasing the awareness of hazardous conditions from the ergonomic perspective and reducing promptly the risk. In order to achieve ergonomic improvements, a real-time analysis of the risk factors combined with a feedback strategy should be implemented. With this regard, particular attention should be dedicated to the selection of an adequate feedback technique depending on the tasks to be performed by the users, environments where the wearable devices are used, and populations wearing the equipment. For instance, Cerqueira et al. [48] underline that auditory signals can be muffled by noisy industrial environments, while visual feedback may limit the users’ field of view. Therefore, further research could be focused on this topic in order to enhance the potentialities of the wearable devices for ergonomic purposes.

### 4.4. Which Ergonomic Criteria Are at the Basis of the Use of These Devices?

Only a minority of the retrieved studies refer to international standards as ergonomic criteria. However, international standards constitute the fundamental building blocks for the development of products, activities, and systems, establishing coherent protocols that can be universally understood and adopted [75]. Furthermore, the standards reflect the views of many stakeholders including technical experts, government representatives, and consumers, and represent traditional documents considered by companies to conduct their activities [76]. In particular, some authors refer to the International Standards ISO 11226 [65] to evaluate the acceptability of static working postures, and ISO 11228-1 [70] to assess manual lifting tasks. Methods listed in the ISO 11228-3 [68] are used by a few studies, and only one employs OCRA, which is the preferred method for detailed risk assessments. In any case, the most frequent focus is on the upper limbs and the trunk, while the lower extremities appear neglected. Finally, other studies do not explicitly state the considered criteria, making the application of the wearable device difficult by other researchers and/or practitioners. It is desirable that an increasing number of smart wearables’ applications refer to standardised criteria and take into account a wide range of parts of the body.

### 4.5. Which Populations Can Benefit from the Use of These Wearable Devices?

Construction workers represent the most frequent population mentioned in the studies. Lee et al. [77] emphasise that “WMSDs are prevalent among construction workers engaged in repetitive motions, heavy lifting, awkward postures, and high-force exertions to perform tasks”, and Palikhe et al. [78] recently warn that “construction is ranked as the most hazardous operation involving musculoskeletal disorders and injuries”. The widespread recognition of the hazardousness related to this working population seems to explain the attention of the retrieved papers about the need to develop a wearable device for personnel involved in construction activities. On the other hand, other working and non-working populations experiencing musculoskeletal disorders are little involved in the studies. Indeed, only one wearable device is designed for health care workers (surgeons), and another one for athletes. This calls for the application of existing smart wearables and the development of new ones for further populations.

### 4.6. Are These Wearable Devices Applied and/or Validated in Real Contexts?

The systematic review shows that a minority of the studies demonstrate the utility of the wearable technology in real contexts. However, feedbacks and advices provided by the users represent valuable pieces of information that cannot be neglected during the design process of smart wearables. For this reason, wider applicability in real contexts and stronger validation would be highly recommended.

### 4.7. Study Limitations and Future Research

Several future research directions have been pointed out above to fill the gaps identified in the literature through our review. However, further work could be undertaken also to overcome the limitations of this study.

Firstly, we analysed the scientific English-language literature available in four electronic databases (i.e., IEEEXplore, PubMed, Scopus, and Web of Science). Therefore, we may have overlooked potential interesting studies if written in a language different from English, or not indexed in the selected electronic databases. Moreover, technical documents and specialised sources (e.g., manufactures and vendors sites) were not searched for. A deep investigation of these possible contributions and future interactions with companies designing and producing such equipment represent one of the main future research directions. The definition of a multidisciplinary study would also permit including a market analysis and the examination of specific technical and engineering factors as comparison dimensions in our framework. Additionally, such multidisciplinary study could allow exploring further themes related to the topic, including social integration, data analytics, data collection and storage.

Another relevant future research direction should deal with design principles of the wearable devices from the ergonomic point of view. Indeed, it would be particularly interesting to analyse the ergonomics of the wearable technology, summarising the state of the art regarding the principles of their design, and identifying the main gaps to be overcome. This will be achievable thanks to an ad hoc literature review involving both scientific and technical literature, and an in-depth investigation of international standards.

Further research could also be dedicated to the potential application of the retrieved wearable devices in particular working environments characterised by limited means for entry or exit, and restricted dimensions, and specific space constraints (e.g., confined spaces [79]). These areas can expose workers to the risk of awkward postures that should be prevented and/or mitigated by means of equipment compatible with the space features. For this reason, the possibility to select non-intrusive wearable technologies or adopt a variable set of sensors depending on the task performed by the workers appears particularly promising and represents an interesting future research topic.

## 5. Conclusions

This paper presents a systematic review of wearable devices proposed for ergonomic purposes in the scientific literature. Through a rigorous three-step selection process, 28 papers containing 24 studies have been identified. We analysed them thanks to eleven comparison dimensions that have been defined for properly answering our primary and secondary research questions. The analysis points out a general interest in developing sensor systems able to acquire data and information in real-time or after users’ exposures, and a large attention on ergonomic physical risk factors and in particular on unfavourable postures. The main results, the highlighted strengths and weaknesses of the different approaches, and the defined framework of analysis could be of interest to both researchers and practitioners. Indeed, they provide researchers with an overview of the state of the art of smart wearables in ergonomics and some insights into the potential future developments of the topic. Additionally, they could support practitioners during the selection of the most suitable wearable technology for ergonomic assessments and improvements in industrial and non-industrial settings.

## Figures and Tables

**Figure 1 sensors-21-00777-f001:**
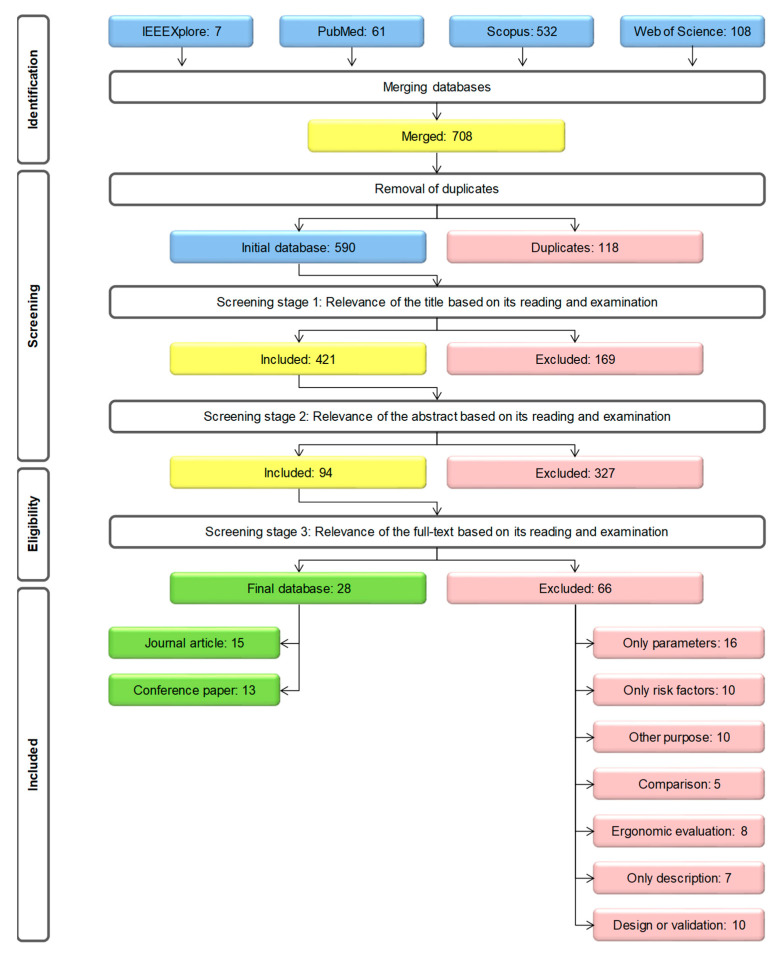
Summary review flow diagram.

**Figure 2 sensors-21-00777-f002:**
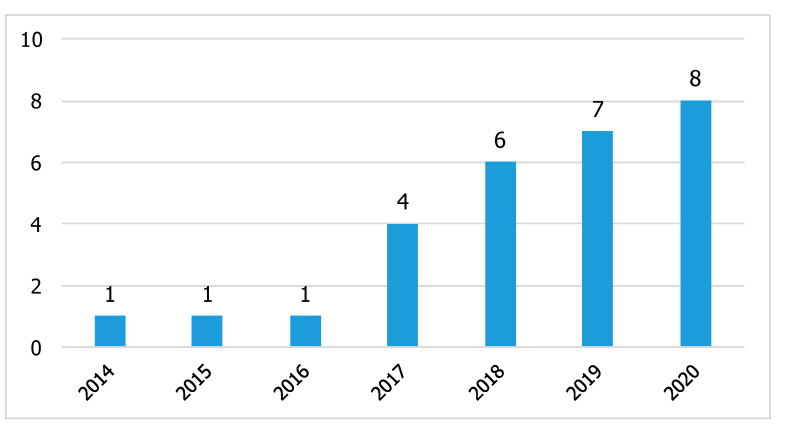
Distribution of papers over time.

**Figure 3 sensors-21-00777-f003:**
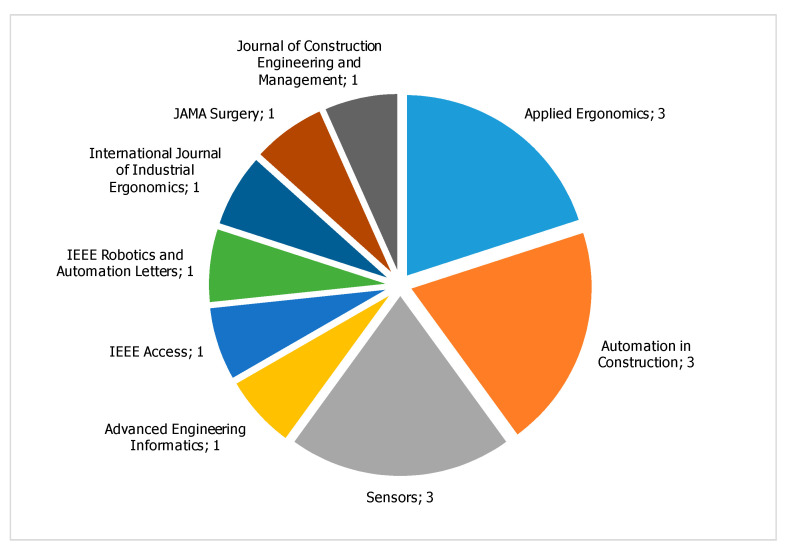
Distribution of the journals that published relevant papers.

**Table 1 sensors-21-00777-t001:** Search strategies for the selected databases.

Database	Search String	Document Type
IEEEXplore	((“Document Title”:“wearable device”) OR (“Abstract”:“wearable device”) OR (“Index Terms”:“wearable device”) OR (“Document Title”:“wearable solution”) OR (“Abstract”:“wearable solution”) OR (“Index Terms”:“wearable solution”) OR (“Document Title”:“wearable system”) OR (“Abstract”:“wearable system”) OR (“Index Terms”:“wearable system”) OR (“Document Title”:“wearable technology”) OR (“Abstract”:“wearable technology”) OR (“Index Terms”:“wearable technology”) OR (“Document Title”:“wearable equipment”) OR (“Abstract”:“wearable equipment”) OR (“Index Terms”:“wearable equipment”) OR (“Document Title”:“wearable computer”) OR (“Abstract”:“wearable computer”) OR (“Index Terms”:“wearable computer”) OR (“Document Title”:“wearable computing”) OR (“Abstract”:“wearable computing”) OR (“Index Terms”:“wearable computing”) OR (“Document Title”:“smart wearable”) OR (“Abstract”:“smart wearable”) OR (“Index Terms”:“smart wearable”)) AND ((“Document Title”:ergonomic*) OR (“Abstract”:ergonomic*) OR (“Index Terms”:ergonomic*) OR (“Document Title”:“human factors”) OR (“Abstract”:“human factors”) OR (“Index Terms”:“human factors”))	Conference paper Journal article
PubMed	((“wearable device”[Title/Abstract] OR “wearable device”[MeSH Terms] OR “wearable solution”[Title/Abstract] OR “wearable solution”[MeSH Terms] OR “wearable system”[Title/Abstract] OR “wearable system”[MeSH Terms] OR “wearable technology”[Title/Abstract] OR “wearable technology”[MeSH Terms] OR “wearable equipment”[Title/Abstract] OR “wearable equipment”[MeSH Terms] OR “wearable computer”[Title/Abstract] OR “wearable computer”[MeSH Terms] OR “wearable computing”[Title/Abstract] OR “wearable computing”[MeSH Terms] OR “smart wearable”[Title/Abstract] OR “smart wearable”[MeSH Terms]) AND (ergonomic*[Title/Abstract] OR ergonomic*[MeSH Terms] OR “human factors”[Title/Abstract] OR “human factors”[MeSH Terms]))	Classical article Congress Journal article
Scopus	TITLE-ABS-KEY((“wearable device” OR “wearable solution” OR “wearable system” OR “wearable technology” OR “wearable equipment” OR “wearable computer” OR “wearable computing” OR “smart wearable”) AND (ergonomic* OR “human factors”))	Article Conference paper
Web of Science	TS=((“wearable device” OR “wearable solution” OR “wearable system” OR “wearable technology” OR “wearable equipment” OR “wearable computer” OR “wearable computing” OR “smart wearable”) AND (ergonomic* OR “human factors”))	Article Proceedings paper

**Table 2 sensors-21-00777-t002:** Relationships between the research questions and comparison dimensions.

Research Question	Comparison Dimension
Which wearable devices have been proposed for ergonomic purposes in the scientific literature?	(1) Wearable device (2) Being ready to wear (3) Part of the body
Which ergonomic risk factors are analysed by means of these wearable devices?	(4) Physical, cognitive, organisational factors (5) Ergonomic risk factor (6) Task
Which ergonomic purposes are achieved by means of these wearable devices?	(7) Main ergonomic purpose and use (8) Output
Which ergonomic criteria are at the basis of the use of these devices?	(9) Ergonomic criteria
Which populations can benefit from the use of these wearable devices?	(10) Population
Are these wearable devices applied and/or validated in real contexts?	(11) Application and validation

**Table 3 sensors-21-00777-t003:** Analysis and comparison of the 24 studies in the reviewed literature.

Study	Wearable Device	Being Ready to Wear	Part of the Body	Ergonomic Risk Factor	Task	Main Ergonomic Purpose and Use	Output	Ergonomic Criteria	Population	Application and Validation
Antwi-Afari et al. [46]	Insole pressure system	Yes	Foot	Posture	Overhead working, squatting, stooping, semi-squatting, and one-legged kneeling	Assessment	Post-exposure recognition of awkward postures using a classification of foot plantar pressure distribution based on a supervised machine learning classifier, using MATLAB^®^	ISO 11226	Workers (construction)	Tests/Simulations
Antwi-Afari et al. [47]	Insole pressure system	Yes	Foot	Physical load	Manual material handling, including holding, carrying, lifting, lowering, pushing, and pulling	Assessment	Post-exposure recognition of overexertion risk using a classification of foot plantar pressure distribution based on a supervised machine learning classifier, using MATLAB^®^	UMass Lowell OSHA	Workers (construction)	Tests/Simulations
Caputo et al. [39,40,41]	Sensor system	No	Upper extremity Trunk Lower extremity	Posture	Assembly task	Assessment	Automatic post-exposure evaluation of static postures by means of an algorithm coded by using MATLAB^®^	EAWS ISO 11226	Workers (industry)	Real contexts Tests/Simulations
Cerqueira et al. [48]	Smart garment	Yes	Upper extremity Trunk	Posture	Tasks requiring awkward postures	Assessment Improvement	Real-time assessment of postures viewable through a GUI created using MATLAB^®^, and biofeedback provided by vibrotactile motors	RULA LUBA	Workers	Tests/Simulations
Conforti et al. [13,42]	Sensor system	No	Trunk Lower extremity Foot	Posture	Manual material handling (e.g., lifting and releasing loads)	Assessment	Post-exposure recognition of awkward postures using a classification of joint angles based on a supervised machine learning classifier, using MATLAB^®^	NIOSH 2007-131 NIOSH 2014-131 NIOSH 94-110	Workers	Tests/Simulations
Conforti et al. [49]	Sensor system	No	Upper extremity Trunk Lower extremity Foot	Physical load	Manual material handling (e.g., lifting and releasing loads)	Assessment	Post-exposure estimation of the forces on the L5/S1 joint using MATLAB^®^	N.A.	Workers	Tests/Simulations
Doshi et al. [50]	Sensor system	No	Upper extremity Trunk	Posture	Driving and manoeuvring vehicles	Assessment	Real-time calculation of the criterion and visualisation by means of an Android application	RULA	Drivers	Tests/Simulations
Ferreira et al. [51]	Smart garment	Yes	Upper extremity Trunk	Posture	Tasks requiring sitting positions	Assessment Improvement	Real-time assessment of the position viewable on a GUI, and feedback realised using luminous signalling (LED)	Principles of ergonomics (proposed in the study)	Workers	Tests/Simulations
Giannini et al. [52]	Sensor system	No	Head Upper extremity Hand Trunk Lower extremity Foot	Physical load	Manual material handling, including lifting and carrying, pushing, pulling, and handling of low loads at high frequency	Assessment	Semiautomatic real-time application of the criteria with results shown on online and offline GUIs	ISO 11228-1/2/3 NIOSH 94-110 Snook & Ciriello method REBA SI	Workers	Real contexts
Hahm and Asada [53]	Robot	Yes	Trunk	Posture	Two-handed manual tasks performed both at and below the floor level	Improvement	Expandable robotic arms with active and passive degrees of freedom to support the user in awkward postures	N.A.	Workers	Tests/Simulations
Jin et al. [54]	Smart- watch	Yes	Hand	Posture Physical load	Application setting, calling, message typing, message checking, and vocal message entry	Improvement	Post-exposure estimation and analysis of joint angles and muscle activity using SAS^®^ and Minitab^®^	N.A.	N.A.	Tests/Simulations
Kim et al. [21]	Feedback interface	No	Upper extremity Trunk Lower extremity	Physical load	Heavy or repetitive manufacturing tasks	Assessment Improvement	Real-time estimation of the overloading joint torque, and vibrotactile feedback by the developed device ErgoTac	Postural risk categories (proposed in the study)	Workers	Tests/Simulations
Kunze et al. [45]	Smart glasses	Yes	Head	Posture Computer vision	Reading and talking	Improvement	Real-time feedback to improve the risk factors, blurring or flipping the screen content away from the user	N.A.	Workers (computer)	Real contexts
Lenzi et al. [55]	Sensor system	No	Upper extremity Trunk	Physical load	Manual handling of low loads at high frequency	Assessment	Post-exposure application of the criteria by means of a software toolbox developed in MATLAB^®^	ISO 11228-3 OCRA Index	Workers	Real contexts
Lins et al. [56]	Sensor system and feedback interface	No	Upper extremity Hand Trunk Lower extremity	Posture	Tasks requiring awkward postures	Assessment Improvement	Real-time recognition of awkward postures using a predefined classifier, and feedback provided by vibrotactile motors	OWAS	Workers (industry)	Tests/Simulations
Lu et al. [57]	Sensor system	No	Upper extremity Hand Trunk Lower extremity	Posture	Two-handed manual lifting	Assessment	Automatic post-exposure recognition of tasks using a machine learning algorithm and estimation of lifting risk variables	NIOSH 94-110 ACGIH TLV for lifting	N.A.	Tests/Simulations
Manjarres et al. [58]	Sensor system	No	Hand Lower extremity	Physical load	Physically demanding jobs and fitness exercises	Assessment	Real-time activity recognition using a classification of heart rate data based on a random forest machine learning classifier, and physical load estimation	Frimat’s criterion	Workers Athletes	Real contexts Tests/Simulations
Meltzer et al. [59]	Sensor system	No	Upper extremity Hand Trunk	Posture	Surgical operating	Assessment	Post-exposure assessment of postures	Posture categories based on RULA	Workers (health care)	Real contexts
Nath et al. [60]	Body- mounted smart- phone	Partly	Upper extremity Trunk	Physical load	Heavy and repetitive activities, including lifting, lowering, carrying, pushing, and pulling	Assessment	Post-exposure recognition of activities using a support vector machine learning classifier, using MATLAB^®^, and risk level estimation	UMass Lowell OSHA	Workers	Tests/Simulations
Nath et al. [61]	Body- mounted smart- phone	Partly	Upper extremity Trunk	Posture	Manual tasks requiring awkward postures	Assessment	Post-exposure assessment of postures and risk level estimation	Postural risk categories (proposed in the study)	Workers (construction)	Tests/Simulations
Peppoloni et al. [43,44]	Sensor system	No	Upper extremity	Posture Physical load	Repetitive tasks (e.g., tasks of supermarket cashiers)	Assessment	Real-time activity segmentation using a state machine-based approach, and application of the criteria with results shown on an online GUI realised using MATLAB^®^	RULA SI	Workers	Real contexts Tests/Simulations
Sedighi Maman et al. [62]	Sensor system	No	Hand Trunk Lower extremity	Physical load	Physically demanding jobs (e.g., assembly tasks, supply pickup and insertion tasks, manual material handling)	Assessment	Post-exposure physical fatigue detection and development modelling	Borg’s Rating of Perceived Exertion	Workers	Tests/Simulations
Valero et al. [63]	Sensor system	No	Upper extremity Trunk Lower extremity	Posture	Bricklaying tasks	Assessment	Post-exposure segmentation of postures using a state machine-based approach, and assessment with results shown on a GUI	ISO 11226	Workers (construction)	Tests/Simulations
Yan et al. [64]	Sensor system	No	Head Upper extremity Trunk	Posture	Manual tasks requiring awkward postures	Assessment Improvement	Real-time estimation of joint angles and assessment of postures, and feedback realised using alarm sounds through a smartphone application	ISO 11226	Workers (construction)	Real contexts Tests/Simulations

Abbreviations: ISO = International Organization for Standardization; UMass Lowell = University of Massachusetts Lowell; OSHA = Occupational Safety and Health Administration; EAWS = European Assembly Work Sheet; GUI = Graphical User Interface; RULA = Rapid Upper Limb Assessment; LUBA = Loading on the Upper Body Assessment; NIOSH = National Institute for Occupational Safety and Health; N.A. = Not Available; LED = Light Emitting Diode; REBA = Rapid Entire Body Assessment; SI = Strain Index; OCRA = Occupational Repetitive Actions; OWAS = Ovako Working Posture Analysis System; ACGIH = American Conference of Governmental Industrial Hygienists; TLV = Threshold Limit Values.

**Table 4 sensors-21-00777-t004:** Number of IMUs and complementary wearable sensors of the available sensor systems.

Study	Number of IMUs	Complementary Wearable Sensors
Caputo et al. [39,40,41]	16	-
Conforti et al. [13,42]	8	-
Conforti et al. [49]	12	2 insoles
Doshi et al. [50]	-	3 flex sensors and 2 gyroscopes
Giannini et al. [52]	17	2 EMG
Lenzi et al. [55]	8	-
Lins et al. [56]	15	-
Lu et al. [57]	5	-
Manjarres et al. [58]	-	1 HR
Meltzer et al. [59]	4	-
Peppoloni et al. [43,44]	3	1 EMG
Sedighi Maman et al. [62]	4	1 HR
Valero et al. [63]	8	-
Yan et al. [64]	2	-

Abbreviations: IMU = Inertial Measurement Unit; EMG = electromyography sensor; HR = heart rate sensor.

## Data Availability

Not applicable.

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
