# Peer review of "Wearable Devices for Ergonomics: A Systematic Literature Review"

_sensors, 2021, doi:10.3390/s21030777_

Round 1

Reviewer 1 Report

The paper organization is in line with the PRISMA guidelines (flow diagram), but the recommended length of a systematic review is more than 20 pages and the proposed paper has 5 lines over 19 pages, among which only 15 have text, the rest being references.

The authors have a large experience in writing systematic literature reviews but in this case the novelty and the added value are not well underlined. How is the authors’ expertise in the area related to the proposed subject?

I think that the  review should be extended by addressing some topics in the area, because wearable devices do not involve only sensors (IMUs especially) or smart garments/glasses/watches: some studies on well-accepted wearable technology, social integration, software backend for wearable technology, data analytics, data collection and storage, all of these for ergonomics of course.

Various manufacturers and vendors sites should be considered for a more technical (engineering) approach and market analysis, although these information are not part of the electronic databases.

In Table 3, the column “Physical, cognitive, organisational factors” can be omitted because for all references only “physical factors” is selected, excepting ref 51 for which cognitive factors is also chosen due to the smart glasses. "Organisational factors" does not appear anymore.

Some keywords must be also omitted: ergonomic assessment vs. ergonomic improvement, manual handling.

Reviewer 2 Report

This work tried to give a review of Wearable devices for ergonomics. Generally, the paper is well written and has good organization. However, the fundamental theory and practical suggestions should be updated in this work. The comments are listed as follows:

  1. The definition of wearable devices is introduced clearly in the introduction. However, the more detail or the fundamental theory about Ergonomics in the introduction is missing. Please give more detail.
  2. There are so many applications in this survey. The reviewer thinks the Ergonomics depends on the requirement of the applications. The author should analyze and categorize the requirement (and purpose) deeply.
  3. More practical and clear suggestions for future studies should be added in this review.
  4. Various topics are discussed in the Discussion. Please add subsections. It can help the reader catch important information.
  5. The sensor placement and number are important. Please give some design principles for ergonomics.

Reviewer 3 Report

The paper reports review on wearable devices for ergonomics applications. The review covered all the areas well but they seem to forget about covering Forcemyography for ergonomics applications. I have the following concerns:

1) Introduction need to be further improved

2) Review of just 28 papers is considered as a narrow and in my opinion dont cover the topic in detail. I suggest authors to consider increasing the numbers.

3) For the above mentioned reasons some of tables and explanations that authors provided seem to be biased.

4) Applications and validation section need to be further improved with more details.

5) Conclusion section is more weak. Bring in more details about the future directions and merge them in conclusion section.

Round 2

Reviewer 1 Report

I consider that the updated version can be published.

Reviewer 2 Report

The authors have tackled the comments. The reviewer suggested acceptance.

Reviewer 3 Report

The authors have addressed all my comments satisfactorily and the paper can be considered for publication.